# IMUMETER—A Convolution Neural Network-Based Sensor for Measurement of Aircraft Ground Performance [note 1]

**DOI:** 10.3390/s21144726

**Published:** 2021-07-10

**Authors:** Jarosław Alexander Pytka, Piotr Budzyński, Paweł Tomiło, Joanna Michałowska, Ernest Gnapowski, Dariusz Błażejczak, Andrzej Łukaszewicz

**Affiliations:** 1Department of Mechanical Engineering, Lublin University of Technology, Nadbystrzycka 36, 20-618 Lublin, Poland; p.budzynski@pollub.pl (P.B.); pawelxtomilo@gmail.com (P.T.); 2The Institute of Technical Sciences and Aviation, The State School of Higher Education, 22-100 Chełm, Poland; jmichalowska@pwszchelm.edu.pl; 3Faculty of Technical Sciences, University College of Administration and Enterprise, 20-150 Lublin, Poland; egnapowski@gmail.com; 4Department of Construction and Usage of Technical Devices, West Pomeranian University of Technology in Szczecin, 70-310 Szczecin, Poland; Dariusz.Blazejczak@zut.edu.pl; 5Faculty of Mechanical Engineering, Bialystok University of Technology, 15-351 Bialystok, Poland; a.lukaszewicz@pb.edu.pl

**Keywords:** aircraft, airfield performance, aircraft motion, artificial neural network, convolution neural network, IMU/GNSS sensor, grass airfield, GARFIELD project

## Abstract

The paper presents the development of the IMUMETER sensor, designed to study the dynamics of aircraft movement, in particular, to measure the ground performance of the aircraft. A motivation of this study was to develop a sensor capable of airplane motion measurement, especially for airfield performance, takeoff and landing. The IMUMETER sensor was designed on the basis of the method of artificial neural networks. The use of a neural network is justified by the fact that the automation of the measurement of the airplane’s ground distance during landing based on acceleration data is possible thanks to the recognition of the touchdown and stopping points, using artificial intelligence. The hardware is based on a single-board computer that works with the inertial navigation platform and a satellite navigation sensor. In the development of the IMUMETER device, original software solutions were developed and tested. The paper describes the development of the Convolution Neural Network, including the learning process based on the measurement results during flight tests of the PZL 104 Wilga 35A aircraft. The ground distance of the test airplane during landing on a grass runway was calculated using the developed neural network model. Additionally included are exemplary measurements of the landing distance of the test airplane during landing on a grass runway. The results obtained in this study can be useful in the development of artificial intelligence-based sensors, especially those for the measurement and analysis of aircraft flight dynamics.

## 1. Introduction

Airfield performance of an airplane plays a significant role in ensuring both safety and comfort during take-off and landing, as well as in terms of aircraft efficiency, including energy savings in motion. The issue was and is the subject of research, which is confirmed by numerous publications, including books in the nature of textbooks or guides [1,2,3,4], but also scientific papers [5,6,7,8,9].

One of the interesting aspects of an airplane’s ground performance is studying the take-off or landing distance of an airplane at a grass airfield. This issue is related to the GARFIELD project, implemented in international cooperation and aimed at creating a publicly available information system on the state of the grass surface of airports and landing strips, and a practical tool such as an online application, whose task is to determine the performance of a given aircraft at a given grass runway [10,11].

The landing and takeoff performance give in the certification specification and further in the Pilot Operating Handbook (POH) are model values, determined for typical conditions (e.g., dry concrete pavement or grass pavement with given soil parameters and grass length, humidity, etc.). Our goal was to develop a sensor that, while being available, cheap and easy to use, will allow the determination of take-off or landing ground roll distances under any conditions, other than those described in the certification documentation. There are still numerous accidents and even catastrophes due to the overestimation of takeoff performance of an airplane on short, unsurfaced runways [12,13,14,15].

Especially when determining the ground performance of light airplanes operating on unsurfaced airfields and opportune grass landing sites, measuring the actual take-off or rollout distance is of great importance. Often, when a pilot arrives at a given landing site, it is unable to determine with sufficient certainty whether the take-off on the way back will be safe, especially if the length of the available runway is small, and the condition of the turf raises additional doubts (long, tall grass, moist, soft soil) [13,14]. The methods of measuring ground performance used in industrial practice are inadequate to the above conditions, mainly due to the difficulties in implementation, the need to use specialized equipment and dependence on technicians making measurements on the ground [16,17].

Rather, it is preferable to use a method based on a simple, lightweight onboard unit. The author of this article made an attempt to use the INS/GNSS (Inertial Navigation System/Global Navigation Satellite System) sensor-based method to determine the ground distances of a PZL 104 Wilga 35A plane, taking off and then landing at the airport with a grass strip. The results of measurements with the use of an RTK (Real-Time Kinematic) device were promising, however, the method suffered from the inconvenience of the need to manually analyze the acceleration of the center of gravity of the aircraft. In addition, professional measuring instruments were used, which were difficult to access, expensive and of relatively large sizes [18,19].

Artificial Intelligence and neural networks offer great opportunities for the recognition of flight phases, which is the basis of the method based on acceleration measurements. Recognition of the various phases of flight is necessary for better training of the neural network. It is important that the neural network be able to recognize flight phases other than those that are required in the analysis of the landing itself (i.e., touchdown and stop), in order to reduce the probability of incorrect phase recognition in the case of, e.g., unusual landing or take-off.

In the problem of recognizing aircraft flight phases, we deal with acceleration time courses, which are of quasi-periodic character. The form and size of the courses are dependent upon flight condition, flight phase and the actual attitude of the airplane. The time courses of acceleration may contain essential information about the corresponding flight configuration, propulsion system function, etc. [20].

Among different deep learning methods, convolution neural networks (CNN) have properties that make them very suitable for recognizing the movement of moving objects. Ignatov, 2018 drew attention to the particularly beneficial effects of using CNN to recognize human activity [20]. Based on this idea, the authors of this study attempted to use convolutional networks to identify phases (states) of aircraft flight. This primary attempt was described in the conference paper [21].

The aim of the present paper was to develop and test a new sensor, based on IMU/GNSS measurement platform, utilizing a CNN for identifying of characteristic phases of flight. The prototype sensor, primarily called IMUMETER, was installed on a test airplane and a series of test flights was conducted in order to examine the proposed method. In this study, we limited our considerations to the landing of an airplane on a grass airfield.

## 2. Design and Development of the IMUMETER Sensor

### 2.1. Aircraft Motion Sensing

The ability to recognize individual phases of an airplane flight brings many benefits and, among other things, may help in the design of a device and method for measuring airplane ground performance. When it comes to landing, the key point is the touchdown, which is the point in time when the landing gear wheels touch the ground. In the case of an airplane with a nose gear, the wheels of the main landing gear touch the ground first. The situation is slightly different in the case of an airplane with a tail wheel, then at the touchdown time, all three wheels should touch the ground simultaneously. A landing technique is also used where the tail wheel touches the ground just prior to the main landing gear wheels.

The moment of touchdown can be recognized on the basis of measurements made with the use of an accelerometer located at any point in the aircraft structure. The acceleration pulse is characterized by very high dynamics, moreover, the observed acceleration is multidirectional. Figure 1 shows the course of vertical acceleration recorded during the landing of the PZL 104 Wilga 35A aircraft. The accelerometer was placed on the floor in the cabin of the aircraft. It is clearly visible the moment when the acceleration amplitude increases rapidly, and the area with the highest amplitude disappears after a time (the airplane breaks off the ground for a moment, jumps up), to then reappear and continue most of the rollout distance [18].

At this point, it is worth taking a closer look at the transition state, shown in detail in the lower graph of Figure 1. This is the transition from the steady state for the approach at which the plane is still in the air and the vertical acceleration a_z_ reaches values of small amplitude. Then, a state is established at which the amplitude of the a_z_ acceleration value becomes significantly greater, as the airplane wheels roll on the grass runway surface, which causes intense vibrations. The transient state lasts about 200 ms and is characteristic for each of the recorded waveforms (each landing). At the touchdown speed of approx. 28 m/s, the airplane travels a distance of approx. 5.6 m during 200 ms.

Likewise, it is possible to recognize the remaining phases of an airplane landing. It may be difficult to recognize the level of flight in relation to the approach and the landing roll. Therefore, in this work, for the recognition of flight phases, it was proposed to use signals from both the accelerometer and the magnetometer, which indicates the position of the object (airplane) in relation to the Earth’s magnetic field lines. Considering the movement of an airplane, it is obvious that in the landing space the lines of the Earth’s magnetic field have a constant direction.

In the cited works [18,19], the recognition and determination of the touchdown point based on the recorded acceleration data were performed manually. The fundamental progress in this study is the application of the method of artificial neural networks to the recognition of particular phases of flight. This solution is based on the method of human activity recognition, as described for example in [22]. Mobile devices, such as smartphones incorporate a number of different sensors such as GPS (Global Positioning System) sensors, audio sensors (i.e., microphones), image sensors (i.e., cameras), light sensors, temperature sensors, direction sensors (i.e., compasses) and accelerometers. Because of substantial computing power, for the ability to interconnect within a global network to send and receive data, a smartphone can be used as a hardware platform for airplane motion sensing.

### 2.2. Convolution Neural Networks

For the purpose of aircraft motion recognition, a convolutional neural network was chosen in this study. CNN is a hierarchical, feed-forward type neural network. The structure of the CNN network is inspired by the system of vision in nature. Characteristic features, compared to standard neural networks, are convolution layers, which occur in addition to fully connected layers. Convolutional layers teach filters that slide along the input data and apply to sub-regions of the input data. The general construction of a CNN-type neural network is described in the following sections [20,23].

#### 2.2.1. Convolutional Layer

In the field of neural networks the convolution is an operation that is based on multiplication weights-filter and input data. In our case, input data is a 2-dimensional (2d) array. So in the case of recognition of a flight phase, a 2d convolution was used. A dot product of input data and filter-sized patch (smaller than input size) results in a single value. An example of a convolution is shown in Figure 2.

In the case of recognition of a flight phase, a 2d convolution was used. Its main parameters are: number of filters, kernel size, stride and padding. Number of filters defines the total number of matrixes with size of the kernel. Kernel describes a filter that will be passed over input data. It applies convolution product on data and can be expressed by Equation (1) [23]:(1)$$G\left(x,y\right)=\omega \times F\left(x,y\right)=\overset{k_{i}}{\underset{\delta x=-k_{i}}{\sum }}\overset{k_{j}}{\underset{\delta y=-k_{j}}{\sum }}\omega \left(\delta x,\delta y\right)\times F\left(x+\delta x,y+\delta y\right)$$
where: ω—kernel; $-k_{i}\leq \;\delta x\leq k_{i},-k_{j}\leq \delta y\leq k_{j}$—elements of kernel.

Stride parameter is responsible for movement kernel over data. Padding defines number of matrix elements that will be added at the sides of the input data. The parameters for this task are: number of filter—24, kernel size—16 × 2, stride—1 and padding—0.

#### 2.2.2. Nonlinearity

Every node in the neural network fully connected layer does a linear operation. For non-linearity, we need to use a non-linear activation function, e.g., rectified linear activation (*ReLu*). This function can be expressed by Equation (2) [23]:(2)$$f\left(x\right)=x^{+}=max\left(0,x\right)$$

#### 2.2.3. Pooling Layer

Features in CNN can be sensitive to the location of elements in input data. To reduce this dependency feature maps need to be downsampled. The task of the operation of max-pooling is to downsample feature maps by determination of the maximum value for each sub-array with a specific size. This pooling layer follows the convolution layer. Taking the average and maximum size of small rectangular data blocks are the two most commonly used algorithms for a pooling layer. The main parameter of this process is pool size, which defines a size data block. In the present study, the size of 3 × 2 was used.

#### 2.2.4. Softmax Layer

The result obtained in the last layer is transferred to the softmax layer, whose task is to determine the probability in the predicted layers, in other words, to work out a classification decision.

Recognition of a flight phase is a classification problem, so for this task, the activation function of the last layer was softmax. This function assumes that each example is a member of exactly one class, it can be expressed by Equation (3) [24]:(3)$$S{\left(y\right)}_{i}=\frac{e^{y_{i}}}{\sum_{j=1}^{n}e^{y_{i}}}$$
where: y—input vector, an input vector that consists of n elements for n classes; $y_{i}$—ith element of the input vector (value between −∞ and +∞); n—number of classes.

#### 2.2.5. Fully-Connected Layer

After all operations, convolution and max-pooling data need to be fed in dense layers (layer with fully connected neurons). The input of the dense layer is a one-dimensional array, so feature maps need to be flattened to this size. In the dense layer, every input is connected and operations are linear with non-linear activation function. A single layer of neurons with soft-max activation function can be expressed by Equation (4):(4)$$f\left(x\right)=S\left(b+W^{T}X\right)$$
where: $W=\left[\begin{array}{c} w_{1}\\ \ldots \\ w_{n} \end{array}\right]$—matrix of weights; $X=\left[\begin{array}{c} x_{1}\\ \ldots \\ x_{n} \end{array}\right]$—matrix of inputs; b—bias.

Weights and biases are estimated by the backpropagation algorithm for each neuron. The weights and biases are based on new inputs. The neural network calculates how new data would have affected the data that was seen previously-stochastic gradient descent. The model in the training process wants to minimalize loss function by finding optimal values for weights and biases. For each step, weights are changed by the value-learning rate in the optimization algorithm.

For flight phase recognition, 2 dense layers with 12 and 1 neuron were used.

For optimization of the neural network, the Adam algorithm was used. It is a stochastic optimization algorithm that only requires first-order gradients. It can be expressed by Equations (5)–(8) [25]:(5)$$\upsilon_{t}=\beta_{1}\times \upsilon_{t-1}-\left(1-\beta_{1}\right)\times g_{t}$$
(6)$$s_{t}=\beta_{2}\times s_{t-1}-\left(1-\beta_{2}\right)\times g_{t}^{2}$$
(7)$$\Delta \omega_{t}=-\eta \frac{\nu_{t}}{\sqrt{s_{t}+\epsilon }}\times g_{t}$$
(8)$$\omega_{t+1}=\omega_{t}+\Delta \omega_{t}$$
where: η—learning rate; $g_{t}$—gradient at time t along ω; $\upsilon_{t}$—exponential average of gradient along ω; $s_{t}$—exponential average of squares of gradients along ω; $\beta_{1}$, $\beta_{2}$—hyperparameters; ϵ—constant for numerical stability.

In this study, the following parameters were approximated experimentally:

η=0.001, $\beta_{1}=0.9$, $\beta_{2}=0.999$, $\epsilon =\left(1\times 10^{-7}\right)$ To calculate model loss, we needed to define a loss function. In this case, along with soft-max activation function in the last layer of the neuron network, we must use a categorical cross-entropy loss. This function can be expressed with the following equation:(9)$$CE=-\overset{C}{\underset{i}{\sum }}t_{i}\log\left(S{\left(y\right)}_{i}\right)$$
where: C—number of classes; $t_{i}$—ground truth vector; $S{\left(y\right)}_{i}$—soft-max function (Equation (2))

### 2.3. Development of the Basic Neural Network

The development of the neural network was divided into two stages. Firstly, we constructed a primary (basic) network in order to examine of the approach is promising. Any neural network should be learned based on a possible big amount of data that possibly best describe the investigated process. For the purpose of learning the network, a laboratory test stand was constructed. The investigated process of airplane landing was divided into three phases, so three classes had to be distinguished during the learning process. The three classes are as below:-approach, with a constant vertical and longitudinal accelerations and slightly inclined position of the object (airplane);-touchdown with a significant peak of vertical acceleration as well as longitudinal acceleration;-rollout with negative longitudinal acceleration, neutral vertical acceleration and neutral vector of magnetic field.

The role of an airplane played a small trolley with a sensor box. As a sensor, a commercially available device, the STI Sensor Tile Box (STMicroelectronics International N.V., Geneva, Switzerland)*,* was used. The classes were determined on the basis of the moment when the trolley with the sensor covered the marker. ARUCO^TM^ (Ava Group of the University of Cordoba, Cordoba, Spain) markers were recognized by means of a miniature video camera, installed on the trolley. It allowed differentiating the length of acceleration measurements in one sample, because the marker could be recognized as obscured at different times.

The network learned in about 3 s for 113 epochs, very quickly. Figure 3 shows results from the learning process of the network. The accuracy after 113 epochs, on a window length of 24 samples, was about 70%. The loss function, shown in Figure 3 middle, also proves good network capabilities. The table of errors (Figure 3 bottom) shows that the accuracy for the approach class was 81.03%, with an error of 8.62% for the touchdown class and 10.34% for the rollout class. The touchdown class achieved an efficiency of 82.02%, with an error in favor of the approach class—7.69% and 10.26% for the rollout class. The rollout class achieved the lowest accuracy of 66.67% with an error of 23.44% for the approach class and 8.89% for the touchdown class.

The network runs on a Raspberry PI^TM^ (Rapsberry Pi Foundation, Cambridge, UK) single-board computer and it is possible to implement the network in the used sensor box. In such a configuration, it is possible to read the prediction result directly.

Due to the construction of the 0-approach and 2-rollout class stations, they have similar feature maps, so they are, to some extent, confused with each other if the trolley is not moving.

The developed network correctly recognizes the individual traffic phases, but due to the overlapping of classes 0 and 2 and due to the small number of classes, false positives occur. One more class could be added that will separate any weird wheelchair movements. False positives may also result from the fact that the network did not receive enough data to learn.

Initially, it was planned to use the so-called transfer learning, which consists of training the network on the way of research in a real object (an airplane). However, based on the results obtained in the process of learning the primary network, it was concluded that a minimum of 50 measurements (50 test flights = landings) will allow distinguishing more classes. Moreover, the time in a single measurement for a given class is longer in the case of a test flight than for the laboratory test stand. So, we decided to develop another (upgraded) neural network, based on flight test results.

## 3. Development and Learning the Advanced CNN

Based on the results obtained during the training of the primary neural network, the development of an advanced network was targeted. It was assumed that for the learning of the advanced network, data collected in flight tests with an airplane would be used. Airplane motion analysis was limited to the landing maneuver and the following flight phases were considered:-approach—the airplane is flying with a nose slightly tilted down, the engine runs at about 30–40% power (a);-flare—this is the phase when the airplane lifts the nose, the flight path is levelled so as to reduce the touchdown speed (b);-touchdown—all the three landing gear wheels touch the ground at the same moment (simultaneously) (c);-roll-out—the airplane is taxiing at decreasing speed (d);-stop (e).

In this way, the studied maneuver was divided into five phases, which at the same time defined the classes in the advanced neural network.

Figure 4 includes three photographs of the test airplane, showing three consecutive flight phases during one landing. A difference between the approach and the flare is clearly visible. It was impossible to recognize these two phases of flight on the basis of indications only from the accelerometer, therefore an additional sensor, a magnetometer, was used.

### 3.1. The Flight Test Experiment

Experimental research was undertaken consisting of flight tests of a selected aircraft, adapted to operate at grass airfields. The PZL 104 Wilga 35A aircraft is a four-seater high-wing aircraft, powered by a 192 kW piston engine. It has a classic undercarriage with a tail wheel. The main landing gear wheels are suspended on trailing arms, which significantly improves ground performance, especially on airports with grass, uneven surfaces.

The flight test campaign took place at Lublin airport (ICAO Code: EPLZ), on a grass runway (the main runway of this airport has a paved surface). Flights were performed in November 2020, in the early afternoon. A total of over 50 flights were carried out, in which four people participated each time (two pilots and two people from among the authors of this paper, carrying out the measurements).

In order to obtain repeatable results, which were necessary for both network training and model validation, flight test measurements were carried out in the fall, early afternoon, when the thermal conditions of the atmosphere were stable, with a steady wind direction roughly opposite to the take-off direction (+/− 20°). Each time, the meteorological conditions provided to the pilot as part of the takeoff clearance were noted by the test engineer conducting the research.

The preparation of the aircraft for the tests consisted of attaching the IMUMETER device (see Figure 5). A sample flight consisted of making a four-turn pattern and landing continued until the plane came to a stop. During the tests, the ground crew carried out measurements of the rollout length using a reference measuring method. Two observers on the ground are assessing the position of the aircraft at the time of touchdown and stopping, relative to established base points. After the measurement, the next flight took off. Figure 6 shows the measurements carried out on the runway.

The sample dataset obtained in the flight test measurements is included in Figure 7 and Figure 8.

Figure 7 shows time courses of the airplane’s accelerations in three orthogonal directions (X-Y-Z), gathered from a sample landing.

As during the fifth phase-stopping, the acceleration values do not change and their values remain the same as at the end of the fourth phase, for the sake of clarity the results are shown without the fifth phase. The data ranges for the individual phases of the flight are indicative (approximate). Determining class affiliation was performed manually. There are some changes in signal, which are characteristic features for a specific class, i.e., rollout class peaks with high amplitude are consequences of ground roughness.

When analyzing the acceleration waveforms, the az acceleration can be seen, which is the most important from the point of view of flight phase recognition, also reaches a high value during the approach. This is probably the result of the impact of vibrations generated by the aircraft engine, the considerable power of which causes such a reaction of the entire aircraft. On the other hand, the highest values of az acceleration were recorded for the touchdown and rollout phases, the difference being not very large compared to the values measured during the approach phase. This emphasizes the need to use a different, additional sensor, thanks to which incorrect recognition of flight phases will be avoided. This additional sensor is the magnetometer.

In Figure 8, the data waveforms recorded by the magnetometer sensors are shown. The most important were the changes in the magnetic field with respect to the vertical (Z) and horizontal (X) axes, as they informed about changes in the position of the aircraft in the vertical plane, along the direction of flight (the inclination of the aircraft resulting from changes in the angle of attack during individual landing phases). Therefore, during the approach phase, the tilt of the plane manifests itself in quite significant values of the magnetic field compared to the flare phase when the airplane takes a horizontal position. The changes in the magnetic field in relation to the transverse axis (Y) are small and could result from the reaction to crosswinds or other disturbing pulses occurring during the flight.

The data shown in both of the figures above represent one exemplary dataset for training the network. In total, about 50 datasets were used to train the network, each of them was obtained during a separate test flight. Therefore, during the test flights, efforts were made to maintain repeatable conditions in terms of the piloting technique and, consequently, the course of the landing.

### 3.2. Data Processing and Creating a Neural Network

The data from the accelerometer were passed through a high-pass filter and then broken down into components. Then, the data were recalculated so that the coordinate system is rotated: the Z axis of the accelerometer corresponds to the axis of gravity. This is due to the fact that the aircraft has three/six degrees of freedom and may be in a configuration in which the accelerometer axis significantly deviates from the vertical direction and, consequently, the measurement results are erroneous. The matrix with the processed acceleration data and the raw data from the magnetometer are convolved and the maximum values are determined. The flatten operation allows you to provide data as input values to the neural network. The network consists of two dense layers (all neurons are connected to each other). Since both layers have the same number of neurons, there is a random shutdown of neurons between them in order to prevent the network from becoming too accustomed to (“dropout”). The first layer of neurons is activated by the *ReLU* function, the second by the soft-max function. Figure 9 shows the structure of the network and Figure 10 shows the structure of the data that pass through the network.

The input data to the neural network is preprocessed accelerometer data and raw magnetometer data from the period of time equals 2 s with a frequency of 30 Hz. So, the input data represent a matrix of size 60 × 6. Value 6 takes from 3 axes from the accelerometer and magnetometer. By further processing all the input data changes, we finally achieve a size equal to the number of classes.

The rotation of the coordinate system allowed to set the Z axis with respect to the gravity axis. This solution allows us to reduce the distribution of components on the other axes of the coordinate system in the event of airplane positions other than horizontal. Rodrigues’ formula was determined by the following equation:(10)vrot=v×cosθ+(k×v)×sinθ+k(k×v)×(1−cosθ)
where: *v*—vector in ℝ3 space; *k*—unit vector defining the rotation axis; *θ*—vector rotation angle *v* with respect to the *k* axis.

### 3.3. Network Learning

The network was taught in 20 epochs, with the following parameters: learning rate = 0.0005, and decay rate = 1 × 10^−6^. The learning results are shown in Figure 11. The accuracy of the network reached the value of 0.935 for the training set and 0.95 for the validation set after 15 epochs, which is a good result. The loss function, both for training and validation, reaches the value of 2 already after five epochs, and it stabilizes after passing 10 epochs. At the 10th learning epoch, there is a local maximum of the loss function, similar to the accuracy where a local decrease is also observed at the 10th epoch. The peak of this kind is a consequence of the use of batch size for the Adam algorithm. The batch dataset for the 10th epoch had to be misaligned for optimization reasons.

The confusion matrix reveals that the lowest degree of confidence occurs in class D—rollout and C—touchdown (80.85% and 86.54%, respectively). In the case of other classes, the degree of confidence is very high, reaching 95% or even more. The lowest values of correct predictions are for classes C and D. In 11.54% of cases, class C is falsely predicted as class B, while class D is predicted as A—8.51% and B—6.38%, and E—4.26%.

The 86.54% accuracy for class C and 11.54% false predictions for class C for B can be explained by the fact that the touchdown phase is significantly shorter than the other classes and therefore data would have to be collected earlier to be sure of the occurrence of characteristic values. This class is assigned to the dataset in the window and thus the initial part of the set was classified as B, hence there was a large share of erroneous predictions for class B.

Summing up, it can be stated that the network learned quite quickly (15 epochs), and the learning results are satisfactory, therefore, it can proceed to the practical application of the developed model for measurements in real conditions.

## 4. Validation of the IMUMETER Sensor

### 4.1. Measurement of the Landing Rollout Distance with the IMUMETER Sensor and a GNSS Sensor

One of the ideas of using the IMUMETER sensor was to use a smartphone as a hardware platform on which the software of the developed neural network is installed. The main point was that the device should be small and accessible to a wide audience, and such requirements were met by a mobile phone with embedded accelerometers and a GNSS sensor. This configuration was tested at this stage, with the magnetic field values still being measured using external sensors.

The basic algorithm for measurement of airplane ground rollout is that the smartphone-based IMUMETER sensor recognizes the moment of touchdown and then gives a trigger input to the distance measuring system, which uses the GNSS sensor that determines the distance based on site coordinates.

It can be assumed that the shape of the Earth is similar to a sphere, so the haversine formula can provide a good approximation of the distance between two points on its surface. The error of such calculation is less than 1% on average. The haversine formula is expressed by the equation below [26]:(11)d=R×c
where: R—average radius of the Earth; c—the result of the following equation:(12)$$C=2\times \arcsin\left(\min\left(1,\sqrt{a}\right)\right)$$
with: a—the result of the equation:(13)$$a=\sin^{2}\left(\frac{\Delta \varphi }{2}\right)+\cos\varphi_{1}\times \cos\varphi_{2}\times \sin^{2}\left(\frac{\Delta \lambda }{2}\right)$$
where: φ—latitude, (indexes: 1—first point, 2—second point); λ—longitude (indexes: 1—first point, 2—second point).

Figure 12 shows the results of a measurement performed during 16 test flights and the measuring method was GPS tracking.

The comparison of the results obtained with the use of the IMUMETER device with the results of the reference method showed significant deviations for most of the repetitions of the measurements. It should also be noted that several repetitions of the measurements with the IMUMETER device were satisfactory (relative error of less than 10%). Based on this observation, it was found that the source of the error was not the neural network method, but the GPS receiver used for the tests. It was a popular receiver built into a Smartphone, whose data acquisition time is more than 1 s.

### 4.2. Measurement of the Landing Rollout Distance with the IMUMETER Sensor and Kalman Filter

Another algorithm for the determination of airplane rollout distance with the use of the IMUMETER sensor is that the neural network recognizes the moments of the touchdown and stop while the distance is calculated based on longitudinal acceleration with the use of the Kalman filter.

In terms of hardware, the IMUMETER sensor is in this case identical to the first solution, but no GPS sensor was needed to determine the route, only an accelerometer.

The distance traveled by airplane is calculated by double integration of the acceleration using a Kalman filter. The main assumption of the algorithm is that the state achieved in time is a consequence of the state at the moment, according to the formula:(14)$$x_{k}=F_{k}x_{k-1}+B_{k}u_{k}+w_{k}$$
with: $F_{k}$—transition matrix; $x_{k-1}$—prior state; $B_{k}$—control/input model; $u_{k}$—control vector; $w_{k}$—process disturbances; k—time moment.

Figure 13 shows the results of measurements using the Kalman filtering method. In this case, the measurement error was significantly smaller, although not acceptable from the point of view of a practical application. One reason for this situation is that the quality of the accelerometer used for the measurements was poor in the sense of in terms of the accuracy of the measurement.

The reference values in Figure 12 and Figure 13 are results of measurements performed on the ground (see Figure 6).

## 5. Conclusions

The paper outlines the development of the IMUMETER sensor for airplane flight dynamics measurements and motion recognition. The sensor is based on artificial intelligence and includes neural network software. The idea of the IMUMETER sensor’s operation is airplane motion phases based on measurements of accelerations and magnetic fields. A prototype of the sensor was built on the Raspberry Pi^TM^ (Rapsberry Pi Foundation, Cambridge, UK) hardware platform with the INS/GNSS sensors platform as well as a three-axis magnetometer. A Convolution Neural Network model was developed to recognize phases of flight. The CNN model was learned with flight data that were collected during a flight test of the PZL 104 Wilga 35A airplane, and an operation from the grass runway of the Lublin Airport. The CNN model software was installed on a smartphone and primary tests were conducted in order to determine landing rollout distance. Results show that although the smartphone-based version of the IMUMETER recognizes airplane motion’s phases correctly, the accuracy of distance measurement was insufficient, mainly due to weak accelerometers and GNSS sensor capabilities (acquisition time 1s). It was concluded that a DGPS-based hardware platform with accelerometers of high-frequency response, such as for example used in the cited works [27,28,29], would be used in an improved version of the IMUMETER, which is being developed as the next step of this project. Additionally, one practical conclusion is that the use of a smartphone as a hardware platform for airplane motion recognition is rather impossible, as a result of too-slow data acquisition related to airplane motion velocity.

One prospective practical use of the IMUMETER sensor would be the “GARFIELD” online information system on runway surface condition as well as airfield performance analysis and prediction for airplanes operation from grass runways.

## Figures and Tables

**Figure 1 sensors-21-04726-f001:**
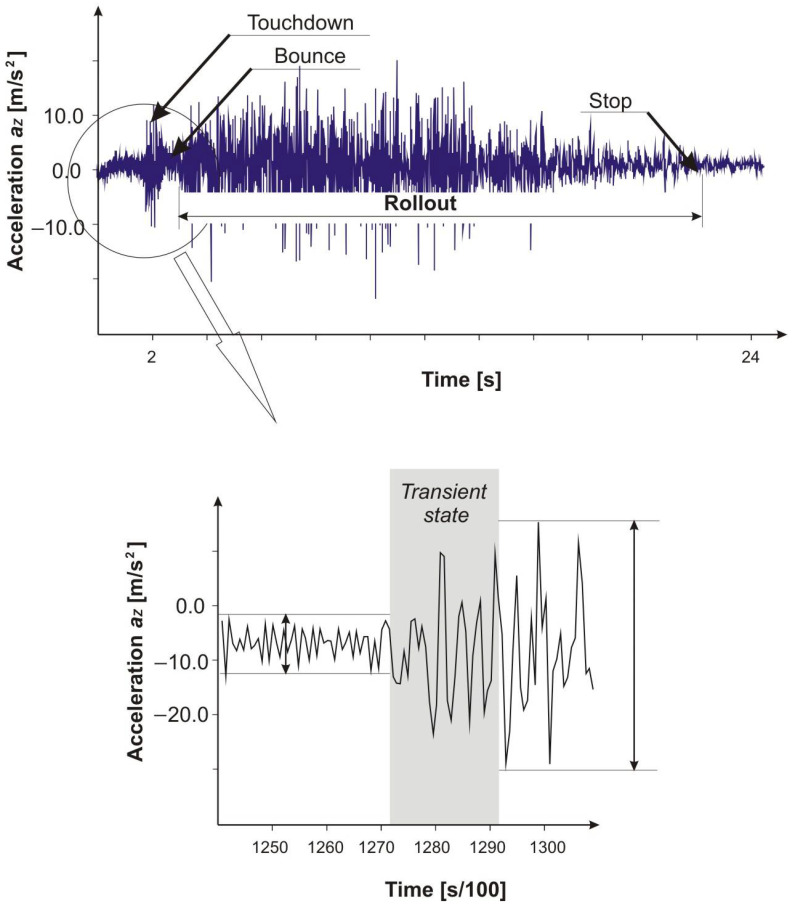
The course of acceleration recorded during the landing of the PZL 104 Wilga 35A aircraft. The apparent moment at which the acceleration amplitude has increased—this is the touchdown time point as the landing gear wheels touch the ground.

**Figure 2 sensors-21-04726-f002:**
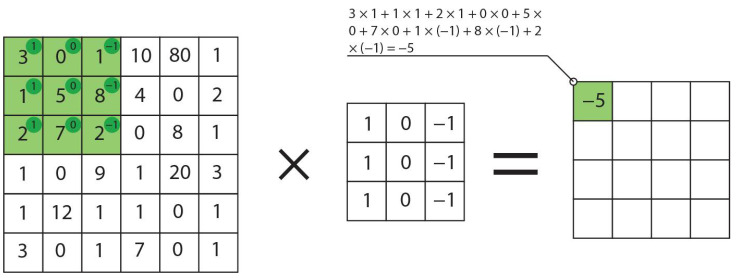
A schematic of convolution operation, obtaining data vector from scalar products of elements of two input data vectors.

**Figure 3 sensors-21-04726-f003:**
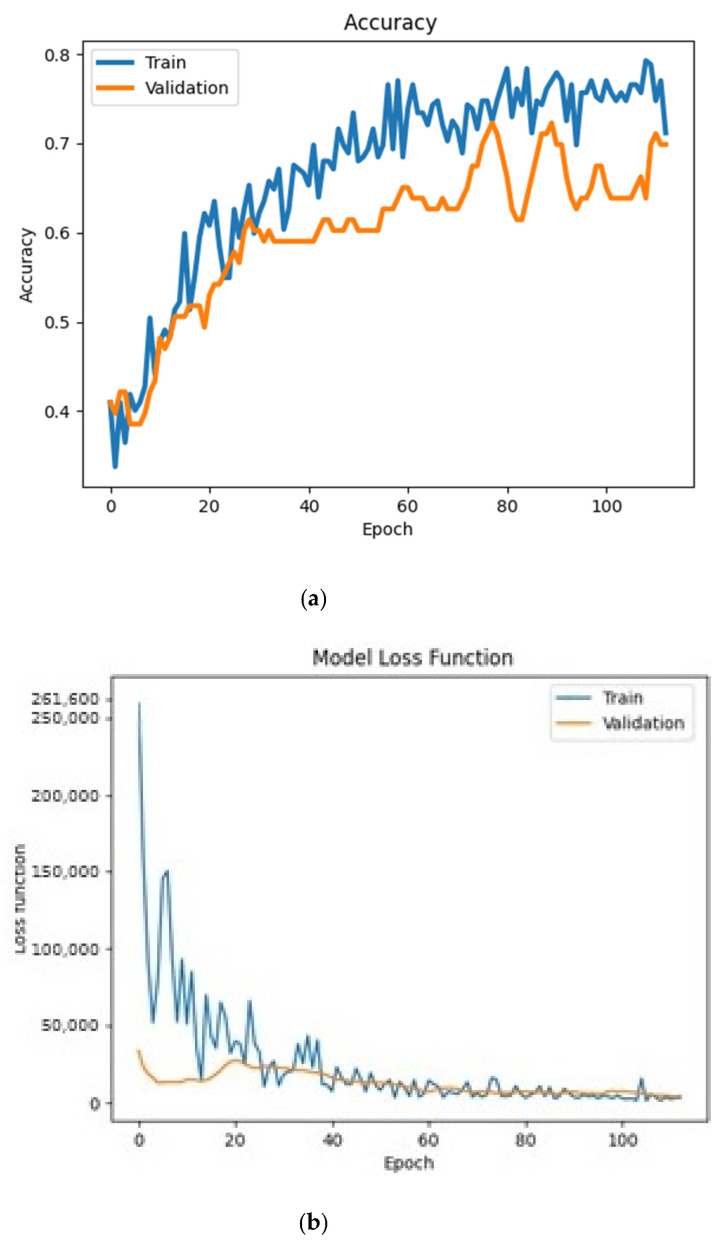
Results of the learning process of the primary neural network. From top to bottom: accuracy of the network, the loss function and the table of errors. (**a**)—approach phase, (**b**)—touchdown, (**c**)—stop.

**Figure 4 sensors-21-04726-f004:**
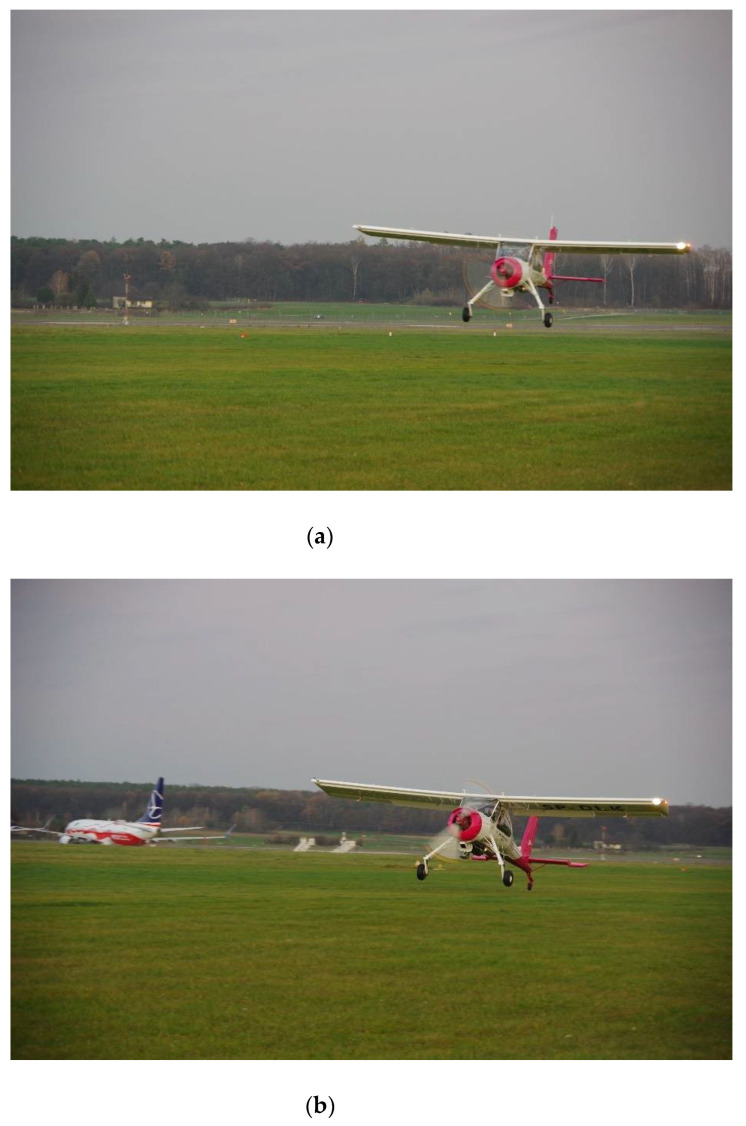
Three consecutive phases of flight during a landing of the test airplane. From top to bottom: (**a**) approach, (**b**) flare and (**c**) touchdown.

**Figure 5 sensors-21-04726-f005:**
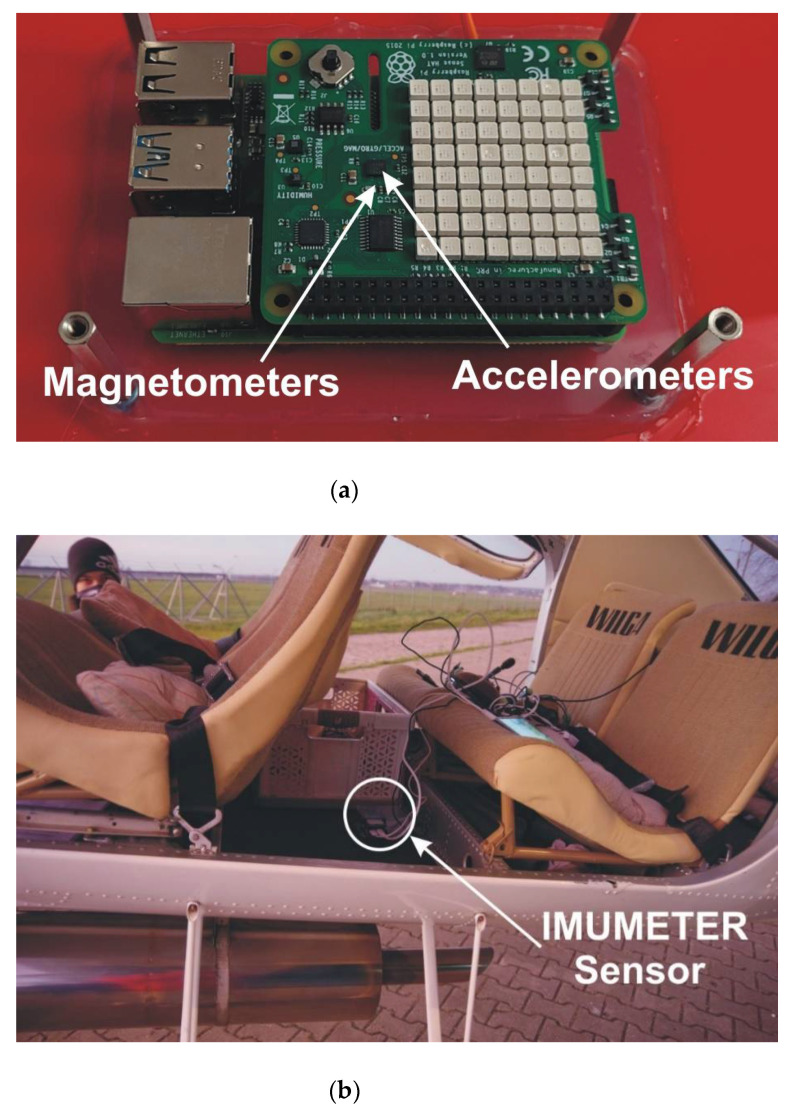
The IMUMETER sensor with accelerometers and magnetometers (**a**) and installation of the sensor in the test airplane (**b**).

**Figure 6 sensors-21-04726-f006:**
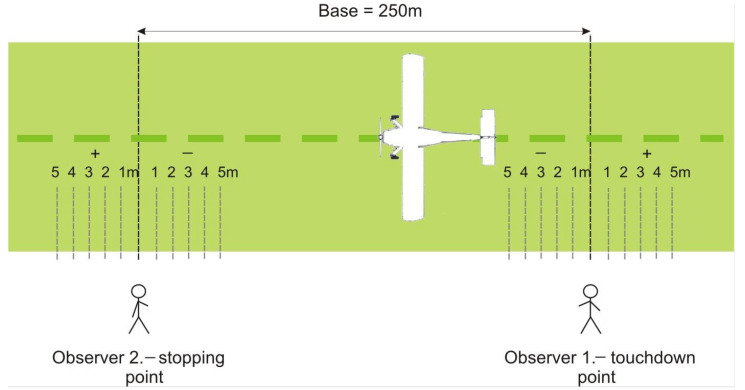
Reference method of measurement of airplane rollout distance on the grass runway.

**Figure 7 sensors-21-04726-f007:**
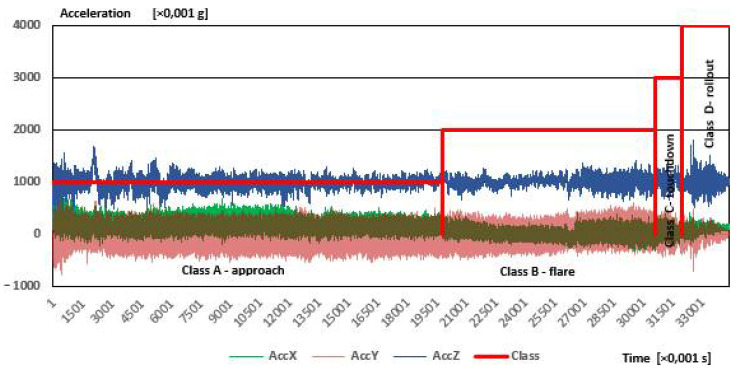
Time courses of accelerations measured during one test landing of the PZL 104 Wilga 35A airplane.

**Figure 8 sensors-21-04726-f008:**
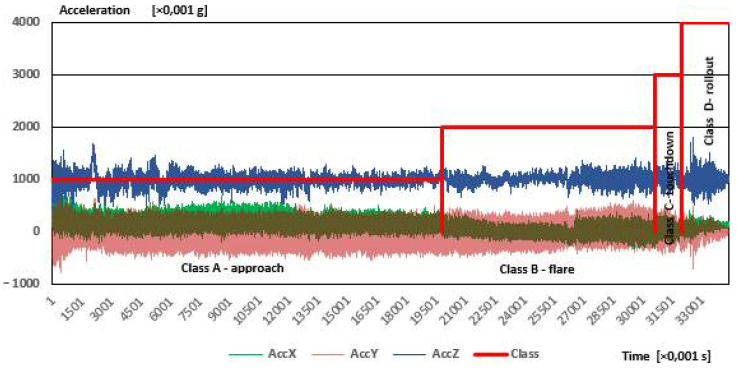
Time courses of magnetic field lines measured during one test landing of the PZL 104 Wilga 35A airplane.

**Figure 9 sensors-21-04726-f009:**
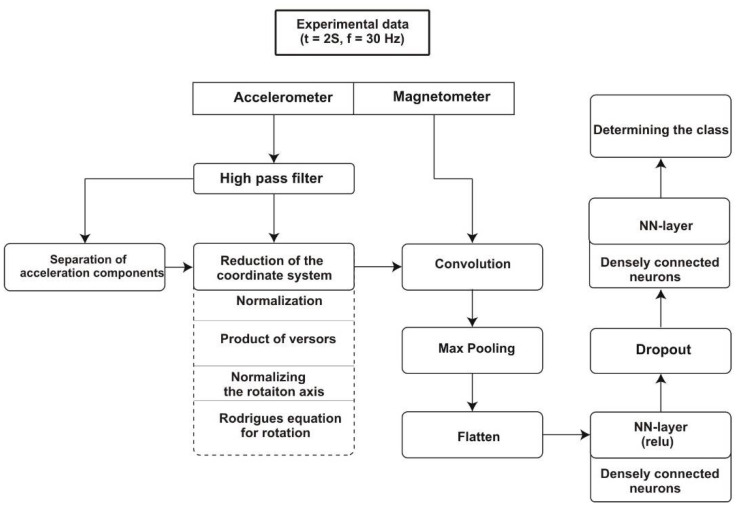
The structure of the convolution neural network.

**Figure 10 sensors-21-04726-f010:**
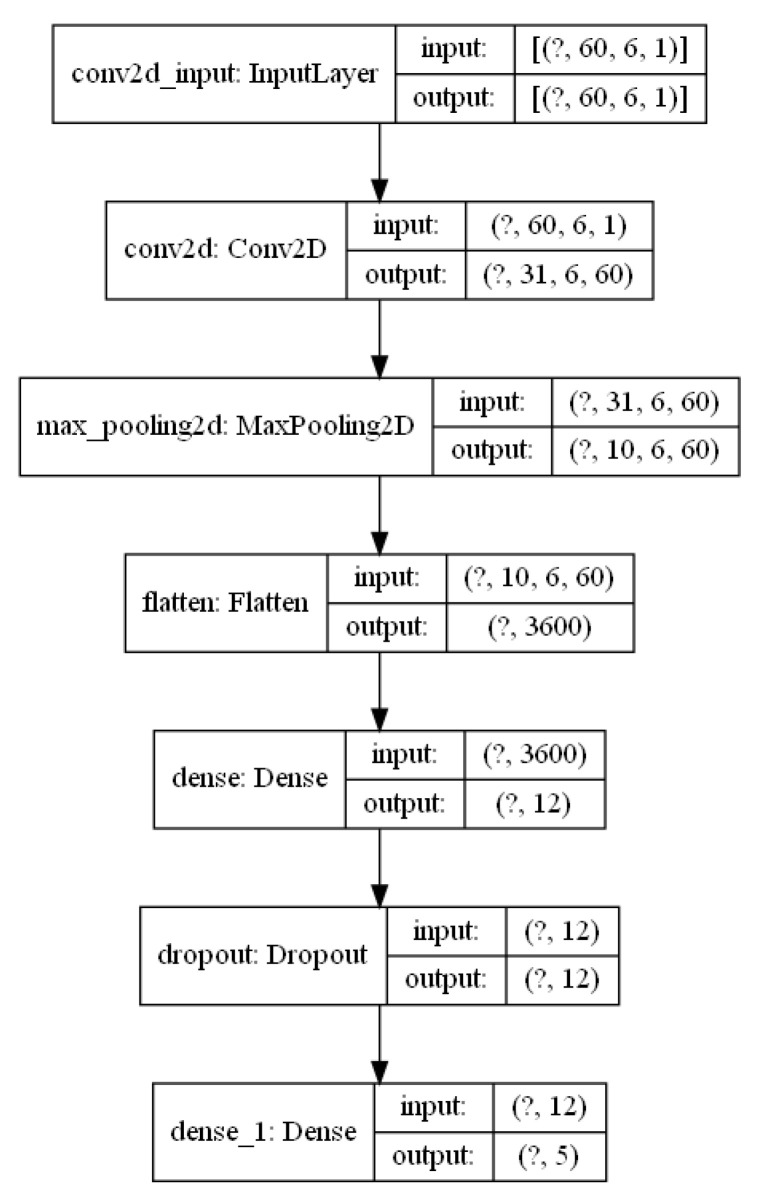
The structure of the data used for network development.

**Figure 11 sensors-21-04726-f011:**
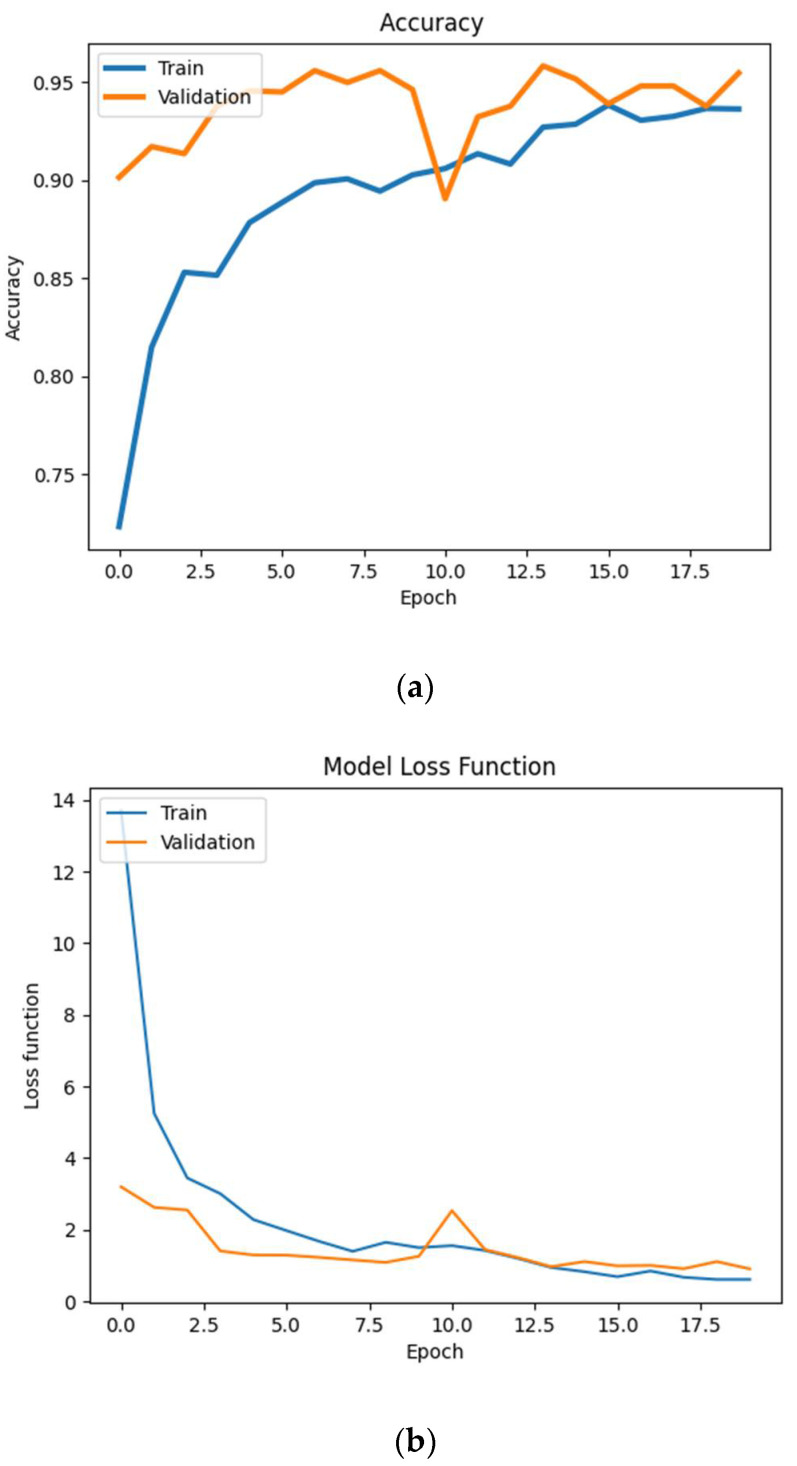
Results of the learning process of the advanced neural network. From top to bottom: (**a**) accuracy of the network, (**b**) the loss function and (**c**) the table of errors. A—approach phase, B—flare, C—touchdown, D—rollout, E—stop.

**Figure 12 sensors-21-04726-f012:**
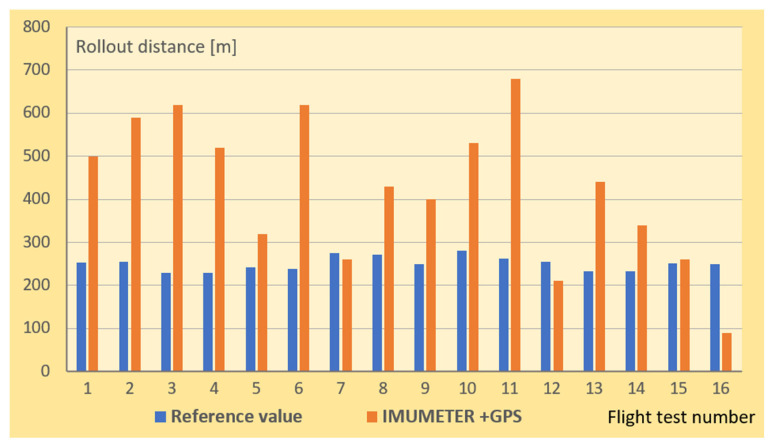
Rollout distance in meters, obtained with the use of the IMUMETER sensor together with a GPS receiver of a smartphone.

**Figure 13 sensors-21-04726-f013:**
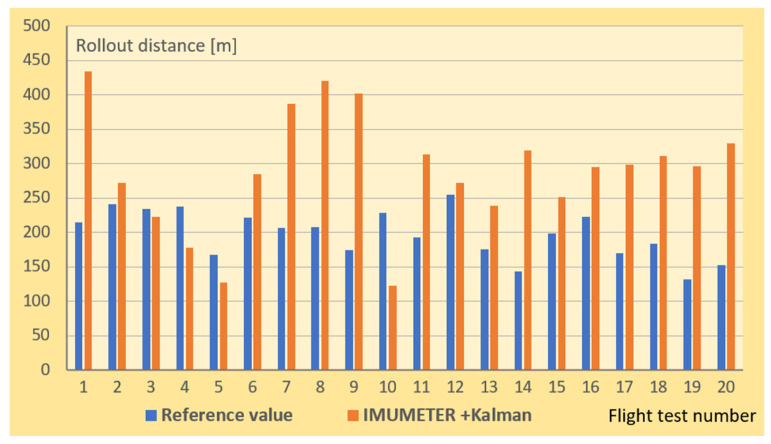
Rollout distance in meters, obtained with the use of the IMUMETER sensor together with the Kalman filtering method.

## Data Availability

Not applicable.

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
