# Peer review of "IMUMETER—A Convolution Neural Network-Based Sensor for Measurement of Aircraft Ground Performance†"

_sensors, 2021, doi:10.3390/s21144726_

Round 1
Reviewer 1 Report
This paper describes development of the Convolution Neural Network, including the learning process based on the measurement results during flight tests of the PZL 104 Wilga 35A aircraft. The topic is interesting and the method is well described. The CNN model software has been installed on a smartphone and primary tests have been conducted in order to determine landing rollout distance. I have some comments here. 1. Figure 7 and 8 contains a lot of information. The author should describe more details. 2. Figure 5 should be improved. 3. How do you select the testing scenarios? Did you consider the different weather conditions?Author Response
Dear Reviewer,
thank you very much for your review. Our response to your comments is included in the attached file.
Best regards,
authors

Reviewer 2 Report
The manuscript is written very interesting and understandable. It gives a good introduction to the method of neural networks. The introduction and motivation is not always well comprehensible. The reference to aviation is not always clearly worked out. Some minor comments below:
Motivation sentence is missing in the abstract.
Explain in the abstract, why the neural network is used
Try to avoid abbreviations in the abstract.
Explain in the abstract, what you explicitly calculated.
Explain the additional value to the research community in the abstract. Explain in the abstract, what the results can be used for.
Line 29: …plays….
Line 30: Try to specify the issue as subject of research. What kind of aircraft performance? Speed, drag, deceleration, navigation accuracy?
Line 41: operating on unsurfaced…
Line: 43: try to avoid he or she. If you do so, consider both he/she
Line 50: It is still not clear, why you want to measure the take-off distance and landing distance of the aircraft. The landing and take-off conditions on different types of surfaces are defined in the certification specification. So, before the aircraft is landing, it should be allowed to do so.
Line 60: Try to make clear, why you must distinguish between flight phases by acceleration when estimating the landing distance.
Figure 1, bottom: please label the arrows (amplitude), please check the x- axis labeling (130000 seconds?),
Figure 1, top: please check the x- axis labeling (24 seconds?), please emphasize the transient phase in the top figure. Please describe the transient phase in the text.
Line 129: …convolution is an operation ….
Figure 2 is very good.
Line 149: … a linear operation …
Equation 2: what does the ^+ mean?
Equation: 4: How do you estimate the weights and the bias?
Figure 3 is very good.
Figures 4 to 6 do not support the scientific process. They may be deleted.
Figure 10 is not very meaningful, because of very abstract data description. What do you want to say with this figure?
Figures 12 and 13: please explain the reference values. Maybe I have missed it in the text, but I don’t understand the origin of the reference values.
Author Response
Dear Reviewer,
thank you very much for your review. We've responded to you comments in the attached file.
With best regards,
authors
